# Large-Scale Answerer in Questioner's Mind for Visual Dialog Question Generation

**Sang-Woo Lee, Tong Gao, Sohee Yang, Jaejun Yoo, & Jung-Woo Ha**
Clova AI Research, NAVER Corp.
{`sang.woo.lee,tong.gao,sh.yang,jaejun.yoo,jungwoo.ha`}@navercorp.com

## Abstract

Answerer in Questioner's Mind (AQM) is an information-theoretic framework that has been recently proposed for task-oriented dialog systems. AQM benefits from asking a question that would maximize the information gain when it is asked. However, due to its intrinsic nature of explicitly calculating the information gain, AQM has a limitation when the solution space is very large. To address this, we propose AQM+ that can deal with a large-scale problem and ask a question that is more coherent to the current context of the dialog. We evaluate our method on GuessWhich, a challenging task-oriented visual dialog problem, where the number of candidate classes is near 10K. Our experimental results and ablation studies show that AQM+ outperforms the state-of-the-art models by a remarkable margin with a reasonable approximation. In particular, the proposed AQM+ reduces more than 60% of error as the dialog proceeds, while the comparative algorithms diminish the error by less than 6%. Based on our results, we argue that AQM+ is a general task-oriented dialog algorithm that can be applied for non-yes-or-no responses.

## 1 Introduction

Recent advances in deep learning have led an end-to-end neural approach to task-oriented dialog problems that can reduce a laborious labeling task on states and intents (Bordes & Weston, 2017). Many researchers have applied sequence-to-sequence models (Vinyals & Le, 2015) that are trained in a supervised learning (SL) and a reinforcement learning (RL) fashion to generate an appropriate sentence for the task. In SL approaches, given the dialog histories so far, the model predicts the distribution of the responses from the task-oriented system (Eric & Manning, 2017; de Vries et al., 2017; Zhao et al., 2018). However, the SL approach typically requires a lot of training data to deal with unseen scenarios and cover all trajectories of the vast action space of dialog systems (Wen et al., 2016). Furthermore, because the SL-based model does not consider the sequential characteristic of the dialog, the error may propagate over time that causes an inconsistent dialog (Li et al., 2017; Zhao & Eskenazi, 2016). To address this issue, RL has been applied to the problem (Strub et al., 2017; Das et al., 2017b). By learning the intrinsic planning policy and the reward function, RL approach enables the models to generate a consistent dialog and generalize better on unseen scenarios. However, these methods struggle to find a competent RNN model that uses back-propagation, owing to the complexity of learning a series of sentences (Lee et al., 2018).

As an alternative, Lee et al. (2018) have recently proposed Answerer in Questioners Mind (AQM) algorithm that does not depend on a limited capacity of RNN models to cover an entire dialog. AQM treats the problems as twenty question games and selects the question that gives a maximum information gain. Unlike the other approaches, AQM benefits from explicitly calculating the posterior distribution and finding a solution analytically. The authors showed promising results in the task-oriented dialog problem, such as GuessWhat (de Vries et al., 2017), where a questioner tries to find an object that is in answerer's mind via a series of Yes/No questions. The candidates are confined to the objects that are presented in the given image (less than ten on average). However, this simplified task may not be general enough to practical problems where the number of objects, questions and answers are typically unrestricted. For example, GuessWhich is a generalized version of GuessWhat that has a greater number of class candidates (9,628 images) and a dialog that

consists of sentences beyond yes or no (Das et al., 2017b). Because the computational complexity vastly increases to explicitly calculate the information gain over the size of the entire search space, the original AQM algorithm is not scalable to a large scale problem. More specifically, the number of the unit calculation for information gain in GuessWhat is 10 (number of objects) $\times$ 2 (Yes/No), while that of GuessWhich is $10,000$ (number of images) $\times \infty$ (answer is a sentence) which makes the computation intractable.

One of the interesting ideas Lee et al. (2018) suggested is to retrieve an appropriate question from the training set. Retrieval-based models, which are basically discriminative models that select a response from a predefined candidate set of system responses, are often used in task-oriented dialog tasks (Bordes & Weston, 2017; Seo et al., 2017a; Liu & Perez, 2017). It is critical not to generate sentences that are ill-structured or irrelevant to the task. However, such a discriminative approach does not fit well with complicated task-oriented visual dialog tasks, because asking an appropriate question considering the visual context is crucial to successfully tackle the problem. It is noticeable that AQM achieved high performance even with a retrieval-based approach in GuessWhat by making the candidate set of questions form the training set. However, Han et al. (2017) pointed out that there exist dominant questions in GuessWhat which can be generally applied to all images (contexts), such as "is it left? or "is it human?. Since GuessWhich is a more complicated task where questions dominant for the game are less likely to exist, it is another reason why the original AQM is difficult to be applied.

To address this, we propose a more generalized version of AQM, dubbed AQM+. Compared to the original AQM, the proposed AQM+ can easily handle the increased number of questions, answers, and candidate classes by employing an approximation based on subset sampling. Particularly, unlike AQM, AQM+ generates candidate questions and answers at every turn, and then selects one of them to ask a question. Because our algorithm considers the previous history of the dialog, AQM+ can generate a more contextual question. To understand the practicality and demonstrate the superior performance of our method, we conduct extensive experiments and quantitative analysis on GuessWhich. Experimental results show that our model could successfully deal with the answers in sentence and significantly decrease 61.5% of the error while the SL and RL methods decrease less than 6% of the error. The ablation study shows that our information gain approximation is reasonable. Increasing the number of sampling by eight times brought only a marginal improvement of percentile mean rank (PMR) from 94.63% to 94.79%, which indicates that our model can effectively approximate the distribution over the large search space with a small number of sampling. Overall, our experimental results provide meaningful insights on how AQM framework can further provide an additional improvement on top of the SL and RL approaches.

Our main contributions are summarized as follows:

- We propose AQM+ that extends the AQM framework toward the more general and complicated tasks. AQM+ can handle a more complicated problem where the number of candidate classes is extremely large.

- At every turn, AQM+ generates a question considering the context of the previous dialog, which is desirable in practice. In particular, AQM+ generates candidate questions and answers at every turn to ask an appropriate question in the context.

- AQM+ outperforms comparative deep learning models by a large margin in Guesswhich, a challenging task-oriented visual dialog task.

## 2  RELATED WORKS

A task-oriented visual dialog problem has recently been paid attention in the field of computer vision and natural language processing (Kim et al., 2017). GuessWhat is one of the famous task-oriented dialog tasks, where the goal is to figure out a target object in the image through a dialog that the answerer has in mind (de Vries et al., 2017). However, GuessWhat is relatively an easy task because it only allows the answer form of yes or no. The baseline visual question answering (VQA) model achieves 78.5%. In the object guessing task (i.e., GuessWhat task itself), the state-of-the-art averaged accuracy of SL, RL (Zhang et al., 2018b), and AQM (Lee et al., 2018) reached 44.6% and 60.8%, and 72.9% at the 5th round, respectively. Random guessing baseline has an accuracy of

Figure 1: Illustration of AQM+ applied for GuessWhich task. The goal of GuessWhich is to figure out a correct answer out of 9,628 test images by asking a sequence of questions.

16.0% (Han et al., 2017), thus RL algorithms achieve 53.3% error decrease, whereas AQM achieves 67.7%.

GuessWhich is a cooperative two-player game that one player tries to figure out an image out of 9,628 that another has in mind (Das et al., 2017b). GuessWhich uses Visual Dialog dataset (Das et al., 2017a) which includes human dialogs on MSCOCO images (Lin et al., 2014) as well as the captions that are generated. Although GuessWhich is similar to GuessWhat, it is more challenging in every task including asking a question, giving an answer, and guessing the target class. For example, unlike GuessWhat that can be answered in yes or no, the answer can be an arbitrary sentence in GuessWhich. Therefore, the VQA task in the Visual Dialog dataset is much studied than the GuessWhat dataset (Lu et al., 2017; Seo et al., 2017b).

Similar to GuessWhat, SL and RL approaches have been applied to solve the GuessWhich task and they showed a moderate increase in performance (Das et al., 2017b; Jain et al., 2018; Zhang et al., 2018a). However, based on the authors' recent Github implementation[1] of the papers in ICCV (Das et al., 2017b), SL and RL methods have shown that only 6% of error is diminished through the dialog compared to the zeroth turn baselines which only use generated caption.

## 3 ALGORITHM: AQM+

### 3.1 PROBLEM SETTING

In our experiments, a questioner bot (Qbot) and an answerer bot (Abot) cooperatively communicate to achieve the goal via natural language. Under the AQM framework, at each turn $t$, Qbot generates an appropriate question $q_t$ and guesses the target class $c$ given a previous history of the dialog $h_{t-1} = (q_{1:t-1}, a_{1:t-1}, h_0)$. Here, $a_t$ is the $t$-th answer and $h_0$ is an initial context that can be obtained before the start of the dialog. We refer to the random variables of target class and the $t$-th answer as $C$ and $A_t$, respectively. Note that the $t$-th question is not a random variable in our information gain calculation. To distinguish from the random variables, we use a bold face for a set notation of target class, question, and answers; i.e. $\mathbf{C}, \mathbf{Q}$, and $\mathbf{A}$.

Figure 1 explains the AQM+ algorithm applied to GuessWhich game. In Figure 1, $c$ is the image with three elephants, $q_1$ is "Are there many people?", $a_1$ is "Yes it is.", $a_2$ is "How many elephants?", and $h_0$ is "There are elephants walking in the zoo." In GuessWhich game, $\mathbf{C}$ is the set of test images whose size is 9,628. The size of $\mathbf{Q}$ and $\mathbf{A}$ is theoretically infinity as questions and answers can be more than one word.

### 3.2 PRELIMINARY: SL, RL, AND AQM APPROACHES

In SL and RL approaches (Das et al., 2017b; Jain et al., 2018; Zhang et al., 2018a), Qbot consists of two RNN modules. One is "Qgen", a question generator finding the solution that maximizes its distribution $p^\dagger$; i.e. $q_t^* = \operatorname{argmax} p^\dagger(q_t|h_{t-1})$. The other is a "Qscore", a class guesser using score function for each class $f^\ddagger(c|h_t)$. Two RNN modules can either be fully separated two RNNs (Strub et al., 2017), or share some recurrent layers but have a different output layer for each (Das et al., 2017b).

On the other hand, in the previous AQM approach (Lee et al., 2018), these two RNN-based models are substituted to the calculation that explicitly finds an analytic solution. It finds a question that

---

[1]https://github.com/batra-mlp-lab/visdial-rl

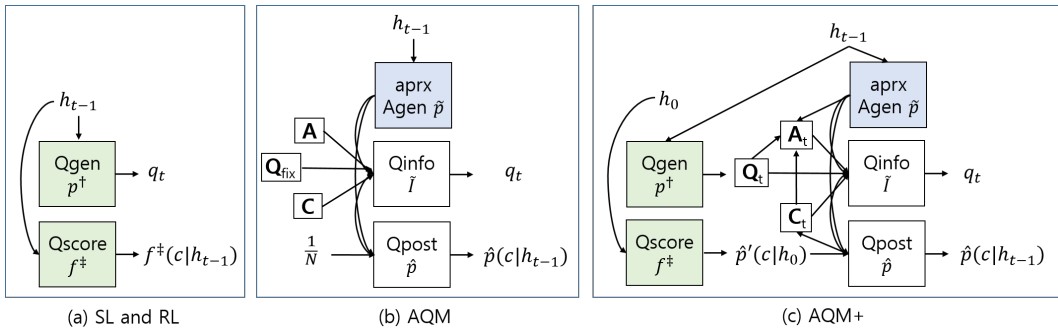

(a) SL and RL  (b) AQM  (c) AQM+

Figure 2: Architecture of AQM+ and comparative models. SL and RL have their main neural modules as Qgen $p^\dagger$ and Qscore $f^\ddagger$, while AQM has aprxAgen $\tilde{p}$ used for Qpost $\hat{p}$ and Qinfo $\tilde{I}$. AQM+ contains all five modules and uses these to make subsets $\mathbf{Q}_t$, $\mathbf{A}_t$, and $\mathbf{C}_t$, thus achieving approximated estimation on information gain for large-scale inference, along with efficient contextual question generation.

Table 1: Notation of Qbot's Modules

| Module | Function | Explanation |
|---|---|---|
| Qgen | $p^\dagger(q_t\|h_{t-1})$ | a question generating RNN |
| Qscore | $f^\ddagger(c\|h_t)$ | a score measuring RNN |
| aprxAgen | $\tilde{p}(a_t\|c, q_t, h_{t-1})$ | an approximated answer generating RNN |
| Qinfo | $\tilde{I}[C, A_t; q_t, h_{t-1}]$ | an information gain calculation function by Equation 1 |
| Qpost | $\hat{p}(c\|h_t)$ | a posterior calculation function by Equation 2 |

Table 2: Notation of Learning Settings

| Learning Setting | Explanation |
|---|---|
| indA | Like SL, aprxAgen is trained from training data |
| depA | Like RL, aprxAgen is trained from the dialog with Abot |
| trueA | aprxAgen shares the parameter with Abot |

maximizes information gain or mutual information $\tilde{I}$, i.e. $q_t^* = \mathrm{argmax}_{q_t \in \mathbf{Q}_{fix}} \tilde{I}[C, A_t; q_t, h_{t-1}]$, where

$$\tilde{I}[C, A_t; q_t, h_{t-1}] = \sum_{c \in \mathbf{C}} \sum_{a_t \in \mathbf{A}} \hat{p}(c|h_{t-1})\tilde{p}(a_t|c, q_t, h_{t-1}) \ln \frac{\tilde{p}(a_t|c, q_t, h_{t-1})}{\tilde{p}'(a_t|q_t, h_{t-1})}. \quad (1)$$

Here, a posterior function $\hat{p}$ can be calculated with a following equation in a sequential way, where $\hat{p}'$ is a prior function given $h_0$.

$$\hat{p}(c|h_t) \propto \hat{p}'(c|h_0) \prod_{j=1}^{t} \tilde{p}(a_j|c, q_j, h_{j-1}) = \hat{p}(c|h_{t-1})\tilde{p}(a_t|c, q_t, h_{t-1}) \quad (2)$$

In AQM, Equation 1 and Equation 2 can be explicitly calculated from the model. For ease of reference, let us name every component one by one. A module that calculates an information gain $\tilde{I}$ is referred to as "Qinfo" and a module that finds an approximated answer distribution $\tilde{p}(a_t|c, q_t, h_{t-1})$ is referred to as "aprxAgen". In AQM, aprxAgen is a model distribution that Qbot has in mind where the target is the true distribution of an answer generator $\bar{p}(a_t|c, q_t, h_{t-1})$, which is referred to as "Agen". Finally, "Qpost" denotes a posterior $\hat{p}$ calculation module for guessing a target class.

As AQM uses full set of $\mathbf{C}$ and $\mathbf{A}$, the complexity depends on the size of $\mathbf{C}$ and $\mathbf{A}$. For the question selection, AQM uses a predefined set of candidate questions ($\mathbf{Q}_{fix}$), which is not changed for a different turn.

## 3.3 AQM+ ALGORITHM

In this paper, we propose AQM+ algorithm, which uses sampling-based approximation, for tackling the large-scale task-oriented dialog problem. The core differences of AQM+ from the previous AQM are summarized as follows:

- The candidate question set $\mathbf{Q}_{t,gen}$ is sampled from $p^{\dagger}(q_t|h_{t-1})$ using a beam search at every turn. Previously, Lee et al. (2018) used a predefined set of candidate questions $\mathbf{Q}_{fix}$. For example, one way to obtain $\mathbf{Q}_{fix}$ is to select questions from the training dataset randomly, called "randQ".

- The answerer model (aprxAgen, $\tilde{p}$) that Qbot has in mind is not a binary classifier (yes/no) but an RNN generator. In addition, aprxAgen does not assume $\tilde{p}(a_t|c, q_t) = \tilde{p}(a_t|c, q_t, h_{t-1})$, which is not even an appropriate assumption when the previous and current questions are sequentially related. For example, $p(a_2 = \text{"yes"} \mid c, q_2 = \text{"is left?"}) \neq p(a_2 = \text{"yes"} \mid c, q_2 = \text{"is left?"}, a_1 = \text{"yes"}, q_1 = \text{"is right?"})$. Regardless of the left term, the probability of the right term is almost zero.

- To approximate the information gain of each question, the subsets of $\mathbf{A}$ and $\mathbf{C}$ are also sampled at every turn. The previous algorithm used full set of $\mathbf{A}$ and $\mathbf{C}$. We describe an additional explanation on our information gain approximation, infogain_topk as below.

**Infogain_topk** The equation for Infogain_topk is as follows:

$$
\tilde{I}_{topk}[C, A_t; q_t, h_{t-1}]
$$
$$
= \sum_{a_t \in \mathbf{A}_{t,topk}(q_t)} \sum_{c \in \mathbf{C}_{t,topk}} \hat{p}_{reg}(c|h_{t-1})\tilde{p}_{reg}(a_t|c, q_t, h_{t-1}) \ln \frac{\tilde{p}_{reg}(a_t|c, q_t, h_{t-1})}{\tilde{p}'_{reg}(a_t|q_t, h_{t-1})}, \quad (3)
$$

where $\hat{p}_{reg}$ and $\tilde{p}_{reg}$ is a normalized version of $\hat{p}$ over $\mathbf{C}_{t,topk}$ and $\tilde{p}$ over $\mathbf{A}_{t,topk}(q_t)$, respectively. Here, $\tilde{p}'_{reg}$ is obtained by using both $\hat{p}_{reg}$ and $\tilde{p}_{reg}$ as follows:

$$
\hat{p}_{reg}(c|h_{t-1}) = \frac{\hat{p}(c|h_{t-1})}{\sum_{c \in \mathbf{C}_{t,topk}} \hat{p}(c|h_{t-1})} \quad (4)
$$

$$
\tilde{p}_{reg}(a_t|c, q_t, h_{t-1}) = \frac{\tilde{p}(a_t|c, q_t, h_{t-1})}{\sum_{a_t \in \mathbf{A}_{t,topk}(q_t)} \tilde{p}(a_t|c, q_t, h_{t-1})} \quad (5)
$$

$$
\tilde{p}'_{reg}(a_t|q_t, h_{t-1}) = \sum_{c \in \mathbf{C}_{t,topk}} \hat{p}_{reg}(c|h_{t-1}) \cdot \tilde{p}_{reg}(a_t|c, q_t, h_{t-1}) \quad (6)
$$

Each set is constructed by the following procedures.

- $\mathbf{C}_{t,topk} \leftarrow$ top-K posterior test images (from Qpost $\hat{p}(c|h_{t-1})$)
- $\mathbf{Q}_{t,gen} \leftarrow$ top-K likelihood questions using the beam search (from Qgen $p^{\dagger}(q_t|h_{t-1})$)
- $\mathbf{A}_{t,topk}(q_t) \leftarrow$ top-1 generated answers from aprxAgen for each question $q_t$ and each class in $\mathbf{C}_{t,topk}$ (from aprxAgen $\tilde{p}(a_t|c, q_t, h_{t-1})$)

Top-K samples may lead our approximation to be biased toward plausible (high-probability) candidate classes and plausible candidate answers. However, we chose to use top-K samples because our main goal is to reduce the entropy over plausible candidate classes and answers, not over the whole candidate classes and answers.

In general, the AQM+ algorithm can deal with various problems where $|\mathbf{C}_{t,topk}|$, $|\mathbf{Q}_{t,gen}|$, and $|\mathbf{A}_{t,topk}(q_t)|$ are all different. Here, $|\cdot|$ denotes the cardinality of a set. We can vary the size of each set and control the complexity of the AQM+ algorithm. In our experiments, however, we mainly considered the problem when $|\mathbf{C}_{t,topk}| = |\mathbf{Q}_{t,gen}| = |\mathbf{A}_{t,topk}(q_t)|$. More specifically, $|\mathbf{C}_{t,topk}|$ is equal to $|\mathbf{A}_{t,topk}(q_t)|$ because our model finds a single best answer $a_t$ given a pair $(q_t, c)$ that maximizes $\tilde{p}(a_t|c, q_t, h_{t-1})$. Therefore, $|\mathbf{A}_{t,topk}| = |\mathbf{Q}_{t,gen}| \cdot |\mathbf{C}_{t,topk}|$ per information

gain calculation where $\mathbf{A}_{t,topk} = \{\mathbf{A}_{t,topk}(q_t) | q_t \in \mathbf{Q}_{t,gen}\}$. For the detailed explanation, see Algorithm 1 in Appendix A.

We also explain the extended sampling method on candidate answers for cases where $\mathbf{A} \neq \mathbf{C}$ is required. In the extended method, aprxAgen first generates top-m answers for each candidate question and each candidate class, where $m$ is the smallest integer satisfying $|A| \leq |C| \cdot m$. After that, the candidate answers are randomly removed, leaving only $|A|$ answers.

## 3.4 LEARNING

In all SL, RL, and AQM frameworks, Qbot needs to be trained to approximate the answer-generating probability distribution of Abot. In AQM approach, aprxAgen does not share the parameters with Agen, and therefore also needs to be trained to approximate Agen. AQM can train aprxAgen by the learning strategy of the SL or RL approach. We explain two learning strategies of AQM framework below: indA and depA. In SL approach, Qgen and Qscore are trained from the training data, which have the same or similar distribution to that of the training data used in training Abot. Likewise, in indA setting of AQM approach, aprxAgen is trained from the training data. In RL approach, Qbot uses dialogs made by the conversation of Qbot and Abot and the result of the game as the objective function (i.e. reward). Likewise, in depA setting of AQM approach, aprxAgen is trained from the questions in the training data and following answers obtained in the conversation between Qbot and Abot. We also use the term trueA, referring to the setting where aprxAgen is the same as Agen, i.e. they share the same parameters. Both the previous AQM algorithm and the proposed AQM+ algorithm use these learning strategies.

## 4 EXPERIMENTS

### 4.1 EXPERIMENTAL SETTING

**GuessWhich Task** GuessWhich is a two player game played by Qbot and Abot. The goal of Guess-Which is to figure out a correct answer out of 9,628 test images by asking a sequence of questions. Abot can see the randomly assigned target image, which is unknown to Qbot. Qbot only observes a caption of the image generated from Neuraltalk2 (Vinyals & Le, 2015). To achieve the goal, Qbot asks a series of questions, to which Abot responds with a sentence.

**Comparative Models** We compare AQM+ with three comparative models, SL-Q, RL-Q, and RL-QA (Das et al., 2017b). In SL-Q, Qbot and Abot are trained separately from the training data. In RL-Q, Qbot is initialized by the Qbot trained by SL-Q and then is fine-tuned by RL. Abot is the same as the Abot trained by SL-Q, and is not fine-tuned further. In the original paper (Das et al., 2017b), it was referred to as Frozen-A. By the way, in an RL-QA setting, not only Qbot but also Abot is concurrently trained with Qbot. In the original paper, it was referred to as RL-full-QAf. We also compare our AQM+ with "Guesser" algorithm. Guesser asks a question generated from SL-Q algorithm and calculates posterior by Qpost of AQM+.

**Non-delta vs. Delta Hyperparameter Setting** The important issue in our GuessWhich experiment is delta setting. In the paper of Das et al. (2017b), SL-Q, RL-Q, and RL-QA algorithms achieve moderate increases of the performance. In SL-Q, 88.5% of percentile mean rank (PMR) is improved to 90.9%. In RL-QA, 90.6% of PMR is improved to 93.3%. Here, 93.3% of PMR at the zeroth turn means that the model can predict the correct image to be more likely than the other 8,983 images out of 9,628 candidates after exploiting the caption information solely. However, Das et al. (2017b) found that another hyperparameter setting, delta, makes much progress on their algorithm. Delta setting refers to different weights on loss and learning decay rate. Based on the authors' recent report on Github, SL-Q and RL-QA methods have shown that less than 6% of error is diminished through the dialog compared to the zeroth turn baseline which only uses generated caption. The PMR of the target (class) image which only uses the caption is around 95.5, but the dialog does not improve the PMR to more than 95.8. We use both non-delta setting (the setting in the original paper) and delta setting (the setting in Github) to test the performance of AQM+.

**Other Experimental Setting** As shown in Figure 2, our model uses five modules, Qgen, Qscore, aprxAgen, Qinfo, and Qpost. We use the same Qgen and Qscore modules as the comparative SL-Q model. In Visual Dialog, Qgen and Qscore share one RNN structure and have different output

Table 3: Test percentile mean rank (PMR) in 10th round. Caption refers the 0th round PMR of SL-Q. The results of comparative deep models in the non-delta setting is from the paper of Das et al. (2017b).

|  | Caption | SL-Q | RL-QA | AQM+ w/ indA | AQM+ w/ depA | AQM+ w/ trueA |
|---|---|---|---|---|---|---|
| non-delta | 88.5 | 90.9 | 93.3 | 94.64 | 97.45 | 99.87 |
| delta | 95.45 | 95.72 | 95.69 | 97.17 | 98.25 | 99.22 |

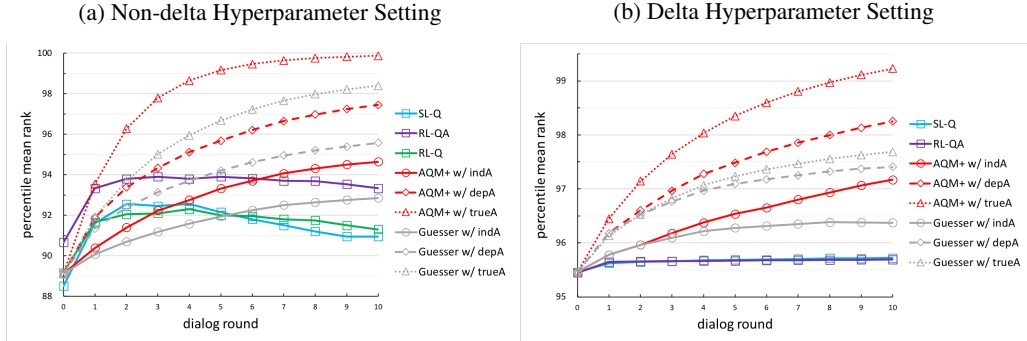

Figure 3: Test percentile mean ranks on GuessWhich experiments.

layers for each. The prior function is obtained from $\hat{p}'(c|h_0) \propto exp(\lambda \cdot f^{\ddagger}(c|h_0))$ using Qscore, where $\lambda$ is a balancing hyperparameter between prior and likelihood. We set $|\mathbf{C}_{t,topk}| = |\mathbf{Q}_{t,gen}| = |\mathbf{A}_{t,topk}(q_t)| = 20$. The epoch for SL-Q is 60. The epoch for RL-Q and RL-QA is 20 for non-delta, and 15 for delta, respectively. Our code is modified from the code of Modhe et al. (2018), and we make our code publically available[2]. All experiments are implemented and fine-tuned with NAVER Smart Machine Learening (NSML) platform (Sung et al., 2017; Kim et al., 2018).

## 4.2 COMPARATIVE RESULTS

Figure 3 shows the PMR of the target image for our AQM+ and comparative models across the rounds. Figure 3a corresponds to the non-delta setting in the original paper (Das et al., 2017b) and Figure 3b corresponds to the delta setting proposed in the Github code.

We see that SL-Q and RL-QA do not significantly improve the performance after a few rounds, especially for the delta setting. In delta setting, SL-Q increases their performance from 95.45% to 95.72% at 10th round, and RL-QA increases their performance from 95.44% to 95.69%. It means that error drop of SL-Q and RL-QA algorithms is 5.74% and 5.33%, respectively. On the other hand, AQM-indA increases its PMR from 95.45% to 96.53% at the fifth round and reaches 97.17% at the 10th round. Likewise, AQM-depA increases its PMR from 95.45% to 97.48% at the fifth round and reach 98.25% at the 10th round, decreasing 61.5% of error. Note that Guesser w/ indA achieves 96.37% at the 10th round, outperforming SL-Q by a significant margin. It shows that not only the question generation but also the guessing mechanism affects the performance degeneration of SL and RL algorithms.

## 4.3 ABLATION STUDY

**No Caption Experiment** We test our AQM+ algorithm where no caption information exists. For the zeroth prediction, we simply replace the prior function from Qscore with a uniform function. Since Qgen in either SL-Q or RL-QA is trained also assuming the existence of the caption, we tried two alternative settings to approximate experiments without a caption. The first trial is the zero-caption experiment, where the caption vector is filled with zeros. The second trial is the random-caption experiment, where the caption vector is replaced with a random caption vector, which is not related to the target image. Figure 4a shows that AQM+ performs well for both zero-caption and random-caption setting. By contrast, SL-Q and RL-QA do not work at all. It seems SL-Q and RL-QA are

---

[2]https://github.com/naver/aqm-plus

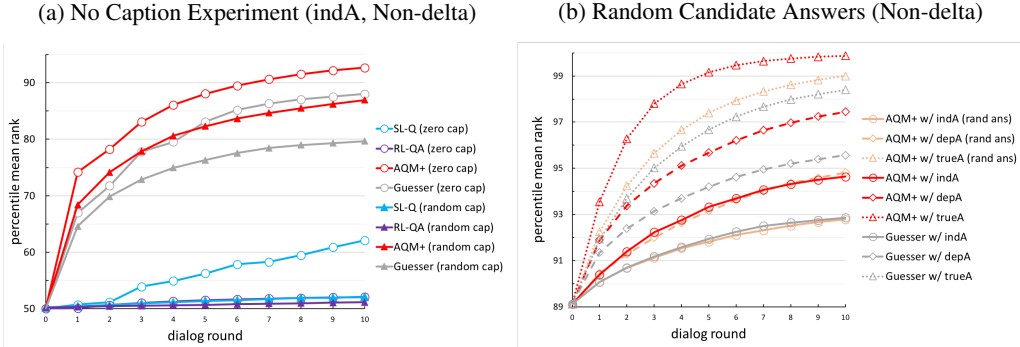

Figure 4: Left column shows the results of ablation studies on no caption experiments. Right column shows the result of ablation studies on random candidate answers experiments, where candidate answers are sampled from the training data.

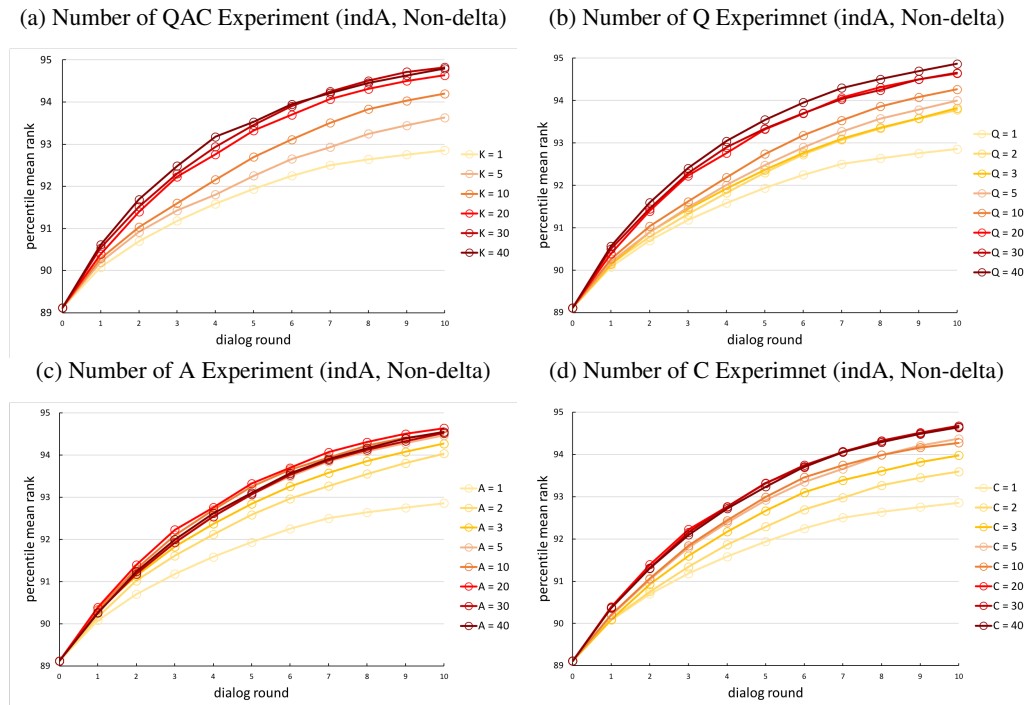

Figure 5: The result of ablation studies on different sizes of the subsets of candidate questions, answers, and classes. In the subfigure (a), the size for three subsets are the same to $K$.

not trained on the situation where zero-caption vector or even totally wrong caption vector comes. Though training SL-Q and RL-QA for these situations can increase their performance, it is evident that SL and RL algorithms are not robust to unexpected environments. Likewise, we also run no caption experiments for depA setting. For more ablation studies, see Figure 7 in Appendix B.

**Random Candidate Answers Experiment** One of our main arguments is that generating candidate questions from Qscore and candidate answers from aprxAgen at every turn makes AQM+ effectively deal with general and complicated task-oriented dialogs. Supporting the argument, we conducted the experiments under the setting where the answer set is randomly selected from the training data and then fixed. Random selection of candidate answers decreases the performance from 94.64% to 92.78% at indA, non-delta, and the 10th round. Appendix B also includes a discussion on the setting with a predefined candidate question set $\mathbf{Q}_{fix}$.

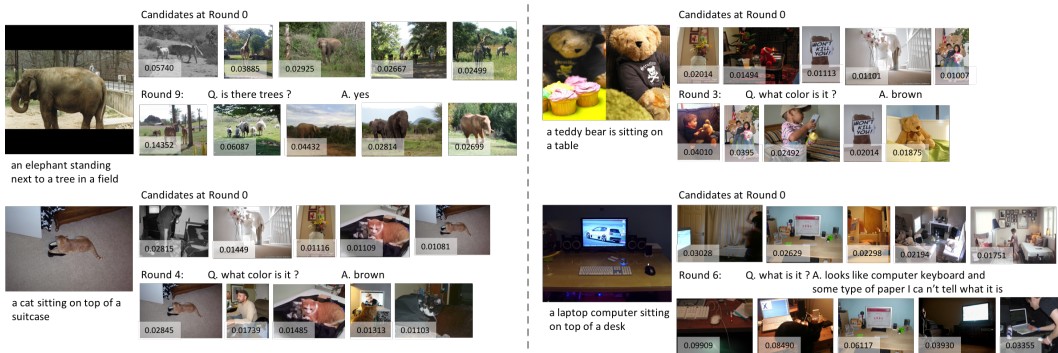

Figure 6: Qualitative results on image retrieval of AQM+. Left column shows true images and their corresponding caption, and right column contains selected top-$k$ images.

**Number of QAC Experiment** We also changed the size of subset $K = |\mathbf{Q}_{t,gen}|=|\mathbf{A}_{t,topk}(q_t)|=|\mathbf{C}_{t,topk}|$ to check our efficiency of information gain approximation, using non-delta setting. Figure 5a shows the experimental results. Note that AQM+ with the setting of $K = 1$ corresponds to Guesser. In the setting of non-delta and indA, 94.64% of PMR is achieved when $K$ is 20, whereas 94.79% is achieved when $K$ is 40. Note that 8 times (2 x 2 x 2) complexity increase just improves 0.15% of PMR, showing the efficiency of the setting of $K$=20 in our experiments. On the other hand, this result also implies that increasing $K$ would make further improvement on the performance. Likewise, in depA setting, changing $K$ from 20 to 40 increases the PMR from 97.44% to 97.77%. For more ablation studies, see Figure 8 in Appendix B. We also changed the size of each subset, $|\mathbf{Q}_{t,gen}|$, $|\mathbf{A}_{t,topk}(q_t)|$, and $|\mathbf{C}_{t,topk}|$. Figure 5b-d shows the results. $|\mathbf{Q}_{t,gen}|$ has the most effect, whereas $|\mathbf{A}_{t,topk}(q_t)|$ has the least effect.

**Generated Questions and Selected Images** Figure 6 shows the top-k images selected by AQM+'s posterior. Non-delta and indA setting is used. The figure shows that relevant images to the caption remained after few dialog turns. The bottom number in the image denotes posterior of the image AQM+ thinks of. We also compare selected examples of generated dialog of SL-Q, RL-QA, and AQM+ w/ indA for delta setting. See Figure 10 in Appendix C for the results.

## 5 DISCUSSION

### 5.1 DIFFICULTY OF GUESSWHICH

According to our results, we infer that PMR degradation of comparative SL and RL models during the dialog is not caused by forgetting dialog context to ask an appropriate question. Comparative results between AQM+ and Guesser show that the improvement from AQM+'s Qpost is significant, which implies that the major constraint of SL and RL is the limited capacity of RNN and its softmax score function.

Another reason for the poor performance lies in the current status of VQA models. According to Das et al. (2017a), they discovered a variety of models, one of which is used in both the study of Das et al. (2017b) and our experiments, and can already reach 41.2% for answer retrieval accuracy from 100 candidate answers, solely using the question without exploiting image and history information. Fully exploiting these factors, however, increases the performance only slightly to 45.5%. As discrimination on different images relies on image and history information, Qbot suffers to gain meaningful information through the dialog. Therefore, applying AQM+ to the GuessWhich problem means that we not only solve a very complicated problem, but also find that the AQM framework is applicable to the situation where the answer has high uncertainty.

### 5.2 NOTES ON COMPARATIVE ANALYSIS

**Fine-tuning both Qbot and Abot through RL** Though RL-QA is the main setting in the work of Das et al. (2017b), there are some reports indicating that fine-tuning both Qbot and Abot is unfair

(de Vries et al., 2017; Han et al., 2017), as one of the ultimate goals in this field is to make a questioner be able to talk with human. If the distribution of Abot is not fixed during RL, Qbot and Abot can make their own language which is not compatible to natural language (Kottur et al., 2017). To prevent this problem, many studies added the objective function of language model during RL (Zhu et al., 2017; Das et al., 2017b). However, even though the generated dialog is tuned to be like human dialog, the performance of RL-QA on the conversation with human would decrease compared to SL-Q, because the distribution of Abot become far from human's (Chattopadhyay et al., 2017; Lee et al., 2018). Moreover, achieving a good performance by fine-tuning both Qbot and Abot is much easier than fine-tuning only Qbot (Zhu et al., 2017; Han et al., 2017). Thus, it is reasonable to compare AQM+ w/ indA and AQM+ w/ depA with SL-Q and RL-Q, respectively.

**Compuational Cost** AQM+ at $K$=20 uses $20\times20\times20$ calculations for information gain. On the other hand, the previous AQM requires $20\times\infty\times9628$ calculations for information gain, which makes the computation intractable. Even if we use only 100 candidate answers, which is in the Visual Dialog dataset (Das et al., 2017a), the previous AQM requires 2500 times as many calculations (20M) as AQM+. On the other hand, AQM+ requires more calculations and thus requires more inference time than SL or RL. AQM+ generates one question within around 3s when $K$=20, whereas SL generates one question within 0.1s. We used Tesla P40 for our experiments. Though the complexity of our information gain is $O(K^3)$, $K$ does not increase the time required for the whole inference in proportion to the cube of K, when $K$=20. It is because calculating the information gain is not the sole resource-intensive part in the whole inference process.

### 5.3 TOWARD PRACTICAL APPLICATIONS

There are plenty of potential future works to improve the performance of AQM+ in real task-oriented dialog applications. For example, robust task-oriented dialog systems are required for appropriately replying to user's questions (Li et al., 2017) and responding for chit-chat style conversation (Zhao et al., 2017). The question quality can also be improved by diverse beam search approaches (Vijayakumar et al., 2016; Li et al., 2016), which prevent sampling similar questions for the candidate set. We highlight two issues described below; online learning and fast inference.

**Online Learning** For a novel answerer, fine-tuning on the dialog model is required (Krause et al., 2018). If the experiences of many users are available, model-agnostic meta learning (MAML) (Finn et al., 2017) can be applied for few-shot learning. Updating the hyperparameter $\lambda$ in an online manner, which balances the effect of the prior and the likelihood, can also be effective in practice. If the answer distribution of user is different from our aprxAgen, we can increase $\lambda$ to decrease the effect of the likelihood.

**Fast Inference** AQM+'s time complexity can be decreased further by changing the structure of aprxAgen. In specific, we can apply diverse methods such as skipping the update of hidden states in some steps (Seo et al., 2018), using convolution networks or self-attention networks (Yu et al., 2018; Vaswani et al., 2017), substituting matrix multiplication operation for hidden state update to weighted addition (Yu & Liu, 2018), and direct information gain inference from the neural networks (Belghazi et al., 2018).

## 6 CONCLUSION

Asking appropriate questions in practical applications has recently been paid attention (Rao & Daumé III, 2018; Buck et al., 2018). We proposed AQM+ algorithm that is a large-scale extension of AQM framework. AQM+ can ask an appropriate question considering the context of the dialog, handle the responses in a sentence form, and efficiently estimate information gain of the target class with a given question. This improvement makes our AQM framework to step forward toward practical task-oriented applications. AQM+ not only outperforms the comparative SL and RL algorithms, but also enlarges the gap between AQM+ and the comparative algorithms comparing to the performance gaps reported in GuessWhat. AQM+ acheives more than 60% error decreases through the dialog, whereas the comparative algorithms only achieve 6% error decreases. Moreover, the performance of AQM+ can be boosted further by employing the models recently proposed in the visual dialog field such as other question generator models (Jain et al., 2018) and question answering models (Kottur et al., 2018).

ACKNOWLEDGMENTS

The authors would like to thank Yu-Jung Heo, Hwiyeol Jo, and Kyunghyun Cho for helpful comments. This work was supported by the Creative Industrial Technology Development Program (10053249) funded by the Ministry of Trade, Industry and Energy (MOTIE, Korea).

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

APPENDIX A. AQM+ ALGORITHM

The question generating process of AQM+ used in our GuessWhich experiments are as follows.

---

**Algorithm 1** Question Generating Process of AQM+ in Our GuessWhich Experiments

---

$\hat{p}'(c|h_0) \propto \exp(\lambda \cdot f^{\ddagger}(c|h_0))$
**for** $t = 1{:}T$ **do**
    $\mathbf{C}_{t,topk} \leftarrow$ top-K posterior test image (from Qpost $\hat{p}(c|h_{t-1})$)
    $\mathbf{Q}_{t,gen} \leftarrow$ top-K likelihood questions using beam search (from Qgen $p^{\dagger}(q_t|h_{t-1})$)
    $\mathbf{A}_{t,topk}(q_t) \leftarrow$ generated answers from aprxAgen for question $q_t$ and each class in $\mathbf{C}_{t,topk}$
    (from aprxAgen $\tilde{p}(a_t|c, q_t, h_{t-1})$)
    $q_t \leftarrow \arg\max_{q'_t \in \mathbf{Q}_{t,gen}} \tilde{I}[C, A_t; q'_t, a_{1:t-1}, q_{1:t-1}]$ with $\mathbf{A}_{t,topk}(q_t)$ and $\mathbf{C}_{t,topk}$ in Eq. 1
    Get $a_t$ from Agen $\bar{p}(a_t|c, q_t, h_{t-1})$
    Update Qpost $\hat{p}(c|h_t) \propto \tilde{p}(a_t|c, q_t, h_{t-1}) \cdot \hat{p}(c|h_{t-1})$ in Eq. 2
**end for**

---

APPENDIX B. ABLATION STUDY

Figure 7 shows the results of the number of QAC ablation experiment on depA and trueA, in the non-delta setting. The effect of K decreases in trueA compared to indA, which indicates that the similarity between the distribution of aprxAgen and Agen is related to the effectiveness of large K. Figure 8 shows the results of the no caption experiment on depA and trueA, in the non-delta setting.

Figure 9 shows the experimental results on the model where AQM+'s Qinfo is used as the question-generator and SL's Qscore is used as the guesser. AQM+s Qinfo does not improve the performance of SLs guesser (Qscore). Our analysis of the results is as follows. For delta setting, the SL guesser is not able to obtain the information from the answers. For the non-delta case, not dialog history but caption information gives dominant information to SLs guesser. The questions which often appear with caption thus gave a more clear signal for the target class for SLs guesser. Figure 9a shows that SL-Q performs better than RL-Q in the early phase, but SL-Qs performance decreases faster than that of RL-Q in the later phase. It is because SL-Q generates the question to be more likely to have co-appeared with the caption than RL-Q. Likewise, AQM+s question does not help SLs guesser because AQM+ generates questions that are more independent of the caption.

We conducted the experiments under the setting where a predefined candidate question set $\mathbf{Q}_{fix}$ is used. The discussion section in the work of Lee et al. (2018) includes an experimental setting in which the candidate questions are generated from an end-to-end SL model only at the first turn. We refer to this setting as gen1Q, as in the previous AQM paper. Figure 10 shows the results of gen1Q ablation study. Note that this setting of $|\mathbf{Q}|{=}100$ requires five times as many computations to calculate the information gain as the original AQM+, despite gen1Q performs even worse than Guesser baseline. Another noticeable phenomenon is that there is no significant performance loss in trueA setting. Since aprxAgen in trueA knows the exact probability of Abot's answer, by exploiting such an aprxAgen, Qbot in trueA can clearly distinguish between different classes by capturing even the subtle differences in answer distributions given similar questions. We also performed the experiments under the setting where $\mathbf{Q}_{fix}$ comes from training data. Figure 11 shows the results of randQ ablation study. The baseline method with this $\mathbf{Q}_{fix}$ showed accuracy degradation. Regardless of the PMR, we point out that randQ retrieves questions relevant to neither the caption nor the target image. It is why we generate candidate questions from a seq-to-seq model.

Figure 12 shows the results of the no history experiment. Dialog history helps to guess the target image but is not critical. Ablating history makes the performance decrease by 0.22% and 0.56% for indA and depA in non-delta, respectively, and 0.46% and 0.21% for indA and depA in delta, respectively.

(a) Number of QAC Experiment (depA, Non-delta)   (b) Number of QAC Experiment (trueA, Non-delta)

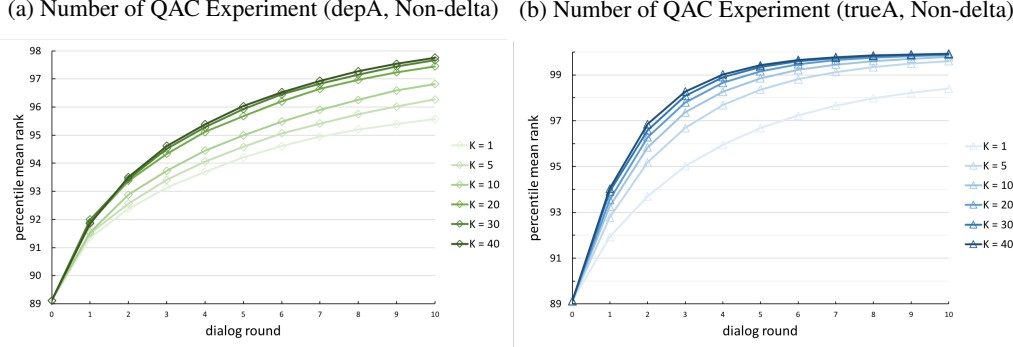

Figure 7: Ablation study on different sizes of the subset of candidate questions, answers, and classes. The size for three subsets are the same to K. The results of the non-delta setting with depA and trueA are illustrated.

(a) No Caption Experiment (depA, Non-delta)   (b) No Caption Experiment (trueA, Non-delta)

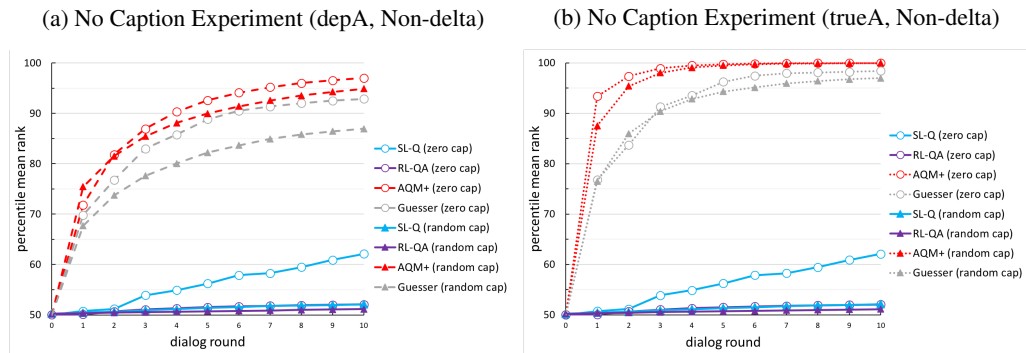

Figure 8: Ablation study on no caption experiment. The results of the non-delta setting with depA and trueA are illustrated.

(a) AQM+'s Qinfo + SL's Qscore (Non-delta)   (b) AQM+'s Qinfo + SL's Qscore (Delta)

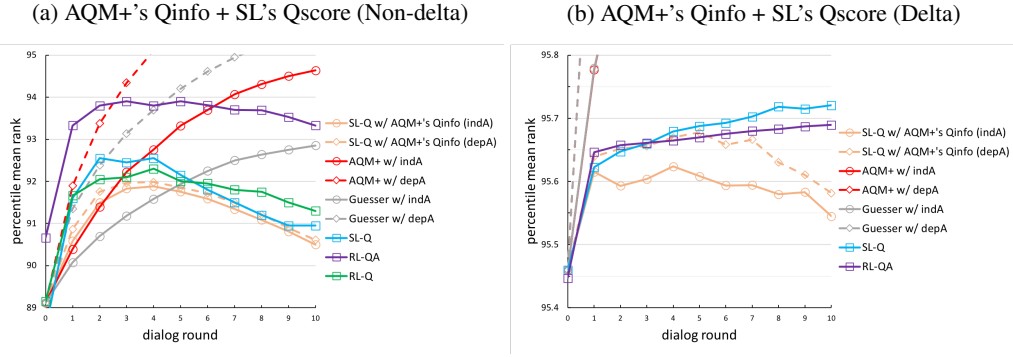

Figure 9: Ablation study on the model with AQM+'s question-generator and SL's guesser.

(a) gen1Q Experiment (Non-delta)   (b) gen1Q Experiment (Delta)

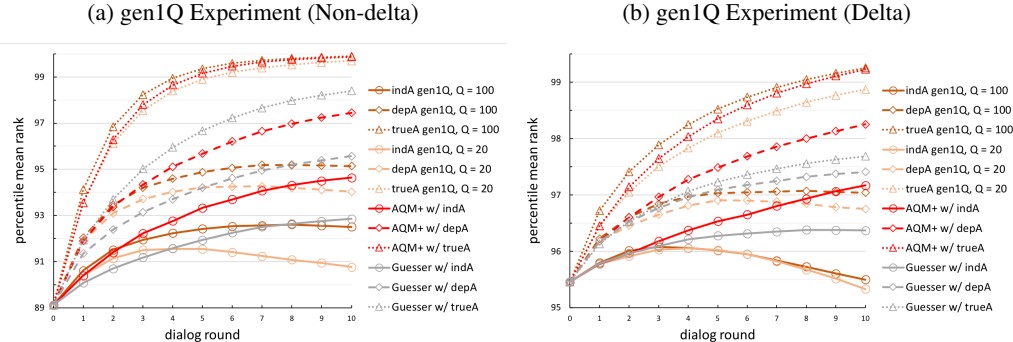

Figure 10: Ablation study on gen1Q. The candidate questions are generated only at the first turn.

(a) randQ Experiment (Non-delta)   (b) randQ Experiment (Delta)

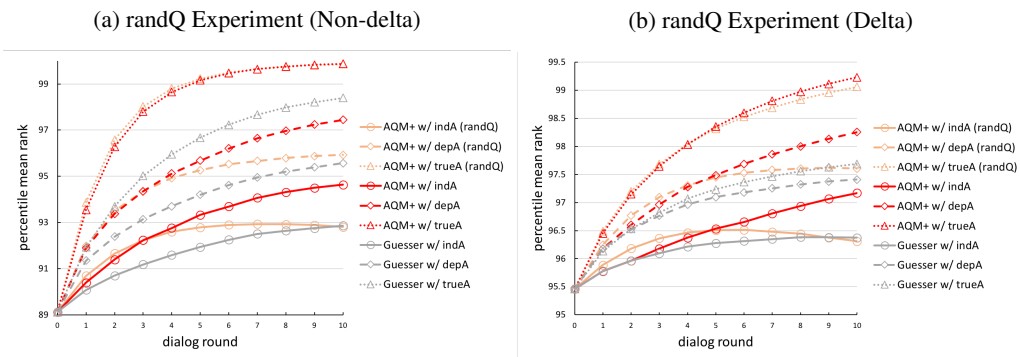

Figure 11: Ablation study on randQ. The candidate questions are extracted from the training data.

(a) No History Experiment (Non-delta)   (b) No History Experiment (Delta)

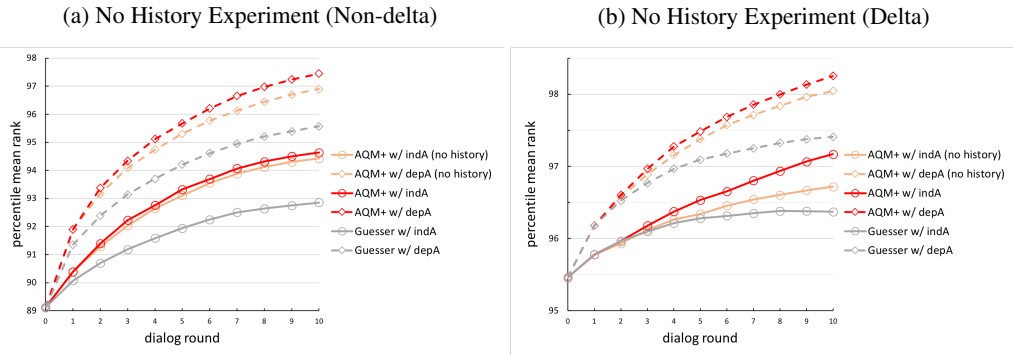

Figure 12: Ablation study on no history experiment. Under this setting, aprxAgen ignores the dialog history.

## Appendix C. Generating Sentences

Figure 13 shows selected examples of generated questions in delta setting. Though delta setting boosts to increase PMR of the zeroth turn much, it degenerates the question quality, especially for RL-QA. Moreover, RL-QA tends to concentrate on the first turn, leaving questions and answers of the remaining turns meaningless.

| Image + Caption | SL-Q | | AQM+ w/ indA | |
|---|---|---|---|---|
| | are there any people ? | no | are there any people? | no |
| | what color of water ? | blue | what kind of bird is it 's body ? | falcon maybe , it 's small |
| | what color is water ? | blue | what color of parrot ? | green |
| | is it sunny ? | yes | are there leaves ? | yes |
| | are there any people ? | yes | what color is bird ? | black and white |
| | are there any other animals | no | what color of water ? | blue |
| | in picture ? | no | does it is water ? | yes |
| | is it 's sky ? | yes | are there trees in ocean ? | yes |
| a bird is standing on a rock in the water | are there clouds ? | ca n't ell | are there grass ? | yes |
| | is bird 's beak ? | yes | can you see sky ? | no |

| | AQM+ w/ indA (gen1Q, Q = 100) | | AQM+ w/ indA (randQ) | |
|---|---|---|---|---|
| | what color is there is there ? | it is black and white | what is man wearing ? | i can only see his shirt which is blue |
| | what color is sky ? | ca n't see it | any words on bus ? | yes , it says UNK |
| | what kind of lake ? | i ca n't tell | is anything in vase ? | yes , it is green |
| | what kind of lake ? | i ca n't tell | can you tell what time of day it is ? | no |
| | what kind of lake ? | i ca n't tell | are cows on farm ? | i ca n't tell , but i think so |
| | what kind of lake ? | i ca n't tell | any words on bus ? | yes , it says UNK |
| | what kind of lake ? | i ca n't tell | what is man wearing ? | i can only see his shirt which is blue |
| | is there grass ? | i ca n't see any | any words on bus ? | yes , it says UNK |
| | what color is stream ? | it 's light brown | is anything in vase ? | no |
| | what kind of water ? | it looks like stream | does man look happy ? | yes |

| | RL-QA | |
|---|---|---|
| | how many birds are there ? | i see 6 seagulls and it would say that it is that it is that caption they are real picture |
| | can you tell what kind of water is ? | no, it is it it is it is it is it is it is it caption caption is |
| | can you tell what kind of water is ? | no , it is it it is it is it it is it is it is that caption |
| | can you tell what kind of water is ? | no , it is it it is it is it is that it is it is it is it caption |
| | can you tell what kind of water is ? | no , it is it it is it is it it is it is it is it caption |
| | can you tell what kind of water is ? | no , it is it it is it is it is it is it is it is it is that it is |
| | can you tell what kind of water is ? | no , it is it it is it is it it is it it is it is it is it is it |
| | can you tell what kind of water is ? | no , it is it it is it is it it is it is it is that it is caption |
| | can you tell what kind of water is ? | no , it is it it is it it is it it is it is that it is it is it |
| | can you tell what kind of water is ? | no , it is it is it is it is it it is caption is that it is that it is that |

| Image + Caption | SL-Q | | AQM+ w/ indA | |
|---|---|---|---|---|
| | is it is box on plate ? | yes | are there see any people ? | no |
| | what  color is plate ? | white | what kind of restaurant ? | ca n't tell |
| | is it large pizza ? | yes | any other than pizza on ? | no |
| | is there appear to cut ? | no | any silverware ? | no |
| | is this look like drinks ? | no | what color of pizza ? | white and green |
| | is this on plate ? | yes | what is on pizza ? | cheese |
| | is there is there ? | no | are there is table ? | yes |
| | is this is there ? | no | is table have table ? | no |
| a pizza sitting on top of a white plate | is there is restaurant ? | no | are there anything else ? | no |
| | is there is pizza ? | yes | what else can you ? | there is pizza in background |

| | AQM+ w/ indA (gen1Q, Q = 100) | | AQM+ w/ indA (randQ) | |
|---|---|---|---|---|
| | are there see any people are there ? | no | is anyone on sofa ? | no |
| | is plate have restaurant ? | i ca n't tell | is it guy or girl ? | it is man |
| | is plate have restaurant ? | i ca n't tell | is floor carpeted ? | no |
| | is plate have restaurant ? | i ca n't tell | are mushrooms sliced thinly or wide piece ? | they are cut into slices |
| | is pizza on table ? | i ca n't tell | can you tell if cake is chocolate ? | it is chocolate frosted donut |
| | is there see pizza in image | no just plate | how does weather look ? | i ca n't really tell |
| | is pizza on table ? | i ca n't tell | is it guy or girl ? | it is boy |
| | is pizza on table ? | i ca n't tell | can you tell if cake is chocolate ? | it is chocolate frosted |
| | is pizza on table ? | i think so | how does weather look ? | it looks like it is sunny |
| | is pizza on table ? | i think so | is it boy or girl ? | it is boy |

| | RL-QA | |
|---|---|---|
| | do you see any people ? | no , just pizza is wrong no people around it is table is caption it is no i would call |
| | can you tell what room they are ? | i ca n't , no it is it it is that it is table it is caption is it caption |
| | can you tell what kind of water is ? | no , it is it it is it is it is it is it it is it caption is that it |
| | can you tell what kind of water is ? | no , it is it it is it is it is it is it is average picture , no , |
| | can you tell what kind of water is ? | no , it is it is it is it is it it is it is that it is |
| | can you tell what kind of water is ? | no , it is it it is it is it is it is no it is it caption caption |
| | can you tell what kind of water is ? | no , it is it it is it is it is it is it is that it is that |
| | can you tell what kind of water is ? | no , it is it is it is it is it is it is it is it is it is it |
| | can you tell what kind of water is ? | no , it is it it is it is it is it is it is it caption is caption |
| | can you tell what kind of water is ? | no , it is it is it is it is it is that it is it it is it is it is |

Figure 13: Selected examples of generated dialog in delta setting.

