# OpenReview forum: "Large-Scale Answerer in Questioner's Mind for Visual Dialog Question Generation"
_ICLR.cc/2019/Conference_

### Official Review · AnonReviewer2 · 2018-10-28

**Rating:** 7
**Confidence:** 5

**Review:**


=================
Updated Thoughts
=================

I was primarily concerned about a lack of analysis regarding the technical contributions moving from AQM to AQM+. The revisions and author comments here have addressed the specific experiments I've asked for and more generally clarified the contributions made as part of AQM+. I've increased my rating to reflect my increased confidence in this paper. Overall, I think this is a good paper and will be interesting to the community.

I also thank the authors for their substantial efforts to revise the paper and address these concerns.


===========
Strengths:
===========

The approach is a sensible application of AQM to the GuessWhich setting and results in significant improvements over existing approaches both in terms of quantitative results and qualitative examples.

===========
Concerns:
===========

[A] Technical Novelty is Limited Compared to AQM
The major departures from the AQM approach claimed in the paper (Section 3.3) are:
	[1] the generation of candidate questions through beam search rather than predefined set
	[2.1] The approximate answerer being an RNN generating free-form language instead of a binary classifier.
	[2.2] Dropping the assumption that \tilde p(a_t | c, q_t) = \tilde p (a_t | c, q_t, h_{t-1}).
	[3] Estimate approximate information gain using subsets of the class and answer space corresponding to the beam-search generated question set and their corresponding answers.

I have some concerns about these:

For [1], the original AQM paper explores this exact setting for GuessWhat in Section 5.2 -- generating the top-100 questions from a pretrained RNN question generator via beam search and ranking them based on information gain. From my understand, this aspect of the approach is not novel.

For [2.1] I disagree that this is a departure from the AQM approach, instead simply an artifact of the experimental setting. The original AQM paper was based in the GuessWhat game in which the answerer could only reply with yes/no/na; however, the method itself is agnostic of this choice. In fact, the detailed algorithm explanation in Appendix A of the AQM paper explicitly discusses the possibility of the answer generator being an RNN model.

Generally, the modifications to AQM largely seem like necessary, straight-forward adjustments to the problem setting of GuessWhich and not algorithmic advances. That said, the changes make sense and do adapt the method to this more complex setting where it performs quite well!


[B] Design decisions are not well justified experimentally
Given that the proposed changes seem rather minor, it would be good to see strong analysis of their effect. Looking back at the claimed difference from AQM, there appear to be a few ablations missing:
- How useful is generating questions? I would have liked to see a comparison to a Q_fix set samples from training. (This corresponds to difference [1] above.)
- How important is dialog history to the aprxAns model? (This corresponds to difference [2.2] above).
- How important is the choice to restrict to |C| classes? Figure 4b begins to study this question but conflates the experiment by simultaneously increasing |Q| and |A|. (This correspond to difference [3] above.)

[C] No evaluation of Visual Dialog metrics
It would be useful to the community to see if this marked improvement in GuessWhich performance also results in improved ability to predict human response to novel dialogs. I (and I imagine many others) would like to see evaluation on the standard Visual Dialog test metrics. If this introspective inference process improves these metrics, it would significantly strengthen the paper!

[D] No discussion of inference time
It would be useful to include discussion of relative inference time. The AQM framework requires substantially more computation than an non-introspective model. Could authors report this relative increase in inference efficiency (say at K=20)?


[E] Lack of Comparison to Base AQM
I would expect explicit comparison to AQM for a model named AQM+ or a discussion on why this is not possible.


===========
Minor Things:
===========

- I don't understand the 2nd claimed contribution from the introduction "At every turn, AQM+ generates a question considering the context of the previous dialog, which is desirable in practice." Is this claim because the aprxAns module uses history?

- Review versions of papers often lack polished writing. I encourage the authors to review their manuscript for future versions with an eye for clarity of terminology, even if it means a departure from established notation in prior work.

- The RL-QA qualitative results, are these from non-delta or delta? Is there a difference between the two in terms of interpretability?

===========
Overview:
===========

The modifications made to adapt AQM to the GuessWhich setting presented here as AQM+ seem to be somewhat minor technical contributions. Further, where these difference could be explored in greater detail, there is a lack of analysis. That said, the proposed approach does make significant qualitative and quantitative improvements in the target problem. I'm fairly on the fence for this paper and look forward to seeing additional analysis and the opinions of other reviewers.

---

> ### Author Response · Authors · 2018-11-17
> **Response to Reviewer 2**
>
> AQM+ generates candidate questions from Qgen and candidate answers from arpxAgen at every turn. This makes AQM+ more efficient and practical for task-oriented dialog systems compared to the base AQM.
>
>
> [A-1]    The original AQM paper explores this exact setting for GuessWhat in Section 5.2 -- generating the top-100 questions from a pretrained RNN question generator via beam search and ranking them based on information gain. From my understand, this aspect of the approach is not novel.
> -    In the recent arXiv manuscript (18/09/21) of Lee et al. (2018), the discussion section includes an experimental setting in which a predefined question set Q_fix consists of the questions generated by an SL-trained seq-to-seq model. Our candidate question generation method in AQM+ can be understood as a future work of the previous paper.
> - To compare the setting of AQM with that of AQM+, we performed an additional ablation study with AQM+ which uses Q_fix generated from a seq2seq model same as the previous AQM. We refer to this setting as gen1Q.
> -    In non-delta and indA setting, gen1Q, the ablated AQM+ model, achieved 90.76% with |Q| = 20 at the 10th round, whereas AQM+ achieved 94.64%at the 10th round.
> -    As in the previous AQM paper, we increased |Q| from 20 to 100. However, gen1Q with |Q| = 100 achieved only 92.50%, which is still lower than the PMR of AQM+. Note that this setting even requires five times as many computations to calculate the information gain as the original AQM+. The result is illustrated in Figure 10.
> -    Our result is similar to the result of Lee et al. (2018). In their report, using a predefined question set generated from a seq2seq model performs slightly better than using a predefined question set extracted from the training data at the 2nd turn (49.79% -> 51.07%, accuracy in GuessWhat), but performs worse at the 5th turn (72.89% -> 70.74%).
>
>
> [A-2]    I disagree that this is a departure from the AQM approach. In fact, the detailed algorithm explanation in Appendix A of the AQM paper explicitly discusses the possibility of the answer generator being an RNN model.
> -    As you mentioned, the previous AQM paper discussed the possibility of using RNN for inferring the distribution of the answer sentence. However, such an extension of approach would be natural and straightforward only in a case where a pre-defined candidate set of answers exists as in Das et al. (2017a). Otherwise, extending the previous AQM approach would not be trivial.
> -    That said, one of the possible natural extensions from AQM to tackle this problem would be to select candidate answers from the training set, like selecting candidate questions in the previous AQM paper. We performed an ablation study on this setting. Random selection of candidate answers decreases the performance from 94.64% to 92.78% at indA, non-delta, and the 10th round. This is because most of the candidate answers are relevant to neither the candidate questions nor the candidate classes. The result is illustrated in Figure 4 (b).
>
>
> [B-0]    Design decisions are not well justified experimentally.
>
> The primary goal of our research is not to find the optimal design for the GuessWhich task, but to make the AQM framework more generalizable and applicable. We would have tried to optimize the model further with other ideas if it were necessary. However, we agree that it is important to conduct a strong analysis on how each of the modifications in AQM+ contributes to the good performance. Therefore, we conducted various ablation studies the reviewer mentioned as explained below.
>
>
> [B-1]    I would have liked to see a comparison to a Q_fix set samples from training.
> -    The ablation study result with a Q_fix randomly extracted from the training data (randQ) showed accuracy degradation (92.79% at indA, non-delta, and the 10th round) compared to the intact AQM+ (94.64%). The result is illustrated in Figure 11.
> Regardless of the PMR, questions retrieved in randQ setting seem to be relevant to neither the caption nor the target image. Figure 13 is revised to include dialogs constructed under this setting.
>
>
> [B-2]    How important is dialog history to the aprxAns model?
> -    Dialog history helps to guess the target image but is not critical. Ablating history makes the performance decrease by 0.22% and 0.56% for indA and depA in non-delta, respectively, and 0.46% and 0.21% for indA and depA in delta, respectively. The results are illustrated in Figure 12.

---

> > ### Comment · AnonReviewer2 · 2018-11-26
> > **Re:Response**
> >
> > Thanks for these updates and paper revisions. I've updated my review above.

---

> > > ### Author Response · Authors · 2018-12-09
> > > **Thank to Reviewer 2**
> > >
> > > Thank you for your consideration and interest in our paper!
> > > We also once again thank you for your comments and valuable suggestions for improving the quality of our paper.

---

> ### Author Response · Authors · 2018-11-17
> **Response to Reviewer 2**
>
> [B-3] How important is the choice to restrict to |C| classes?
> -    In the case of |A| = |C|, we sample candidate answers by generating a top-1 answer from aprxAgen for each candidate question and each candidate class. However, our submitted manuscript does not explain the case where |A| != |C|. For the ablation study on |C|, we extended our sampling method on candidate answers to cover such a case and explained it in the AQM+ subsection (Subsection 3.3).
> -    We conducted the ablation studies on |Q|, |A|, and |C|. In the ablation study on |C|, we changed |C| but fixed |Q|=|A|=20. |Q| has the most effect, whereas |A| has the least effect. Figure 5 (b-d) describes the experimental result in the ablation study subsection.
>
>
> [C]    No evaluation of Visual Dialog Metrics.
> -    As discussed in the reply to AnonR1, we used the same Abot model as that of Das et al. (2017a), and thus the performance on the Visual Dialog Metrics that the reviewer asked for would be the same as the one reported in Das et al. (2017b). There is no metric suggested for Qbot in the paper of Das et al. (2017a). Please let us know if this does not seem to address the point you made.
>
>
> [D]    No discussion of inference time
> -    AQM+ generates one question within around 3s at K=20, whereas SL generates one question within 0.1s. We used Tesla P40 for our experiments. Though the complexity of our information gain is O(K^3), K does not increase the time required for the whole inference in proportion to the cube of K, when K = 20. It is because calculating the information gain is not the sole resource-intensive part in the whole inference process and we do parallel processing for the calculation of aprxAgen using GPU. We did not fully optimize the inference time of AQM+ yet, and the inference time would further decrease if more parallel processing techniques are applied. We added the description on this issue in the discussion section.
>
>
> [E]    Lack of Comparison to Base AQM
> -    The main setting of AQM+ in the paper uses 20x20x20 calculations for information gain. On the other hand, the base AQM requires 20 x infinity x 10000 calculations for information gain, which makes the computation of the base AQM intractable. Even if we have 100 candidate answers as in Visual Dialog (Das et al. (2017a)), the base AQM requires 2500 times as many calculations (20M) as AQM+. We added the description on this issue in the discussion section.
> -    On the other hand, we conducted extensive ablation studies to indirectly compare the base AQM with our AQM+, as explained above.
>
>
>
> Minor Things:
>
>
> [Minor1]     I don't understand the 2nd claimed contribution from the introduction "At every turn, AQM+ generates a question considering the context of the previous dialog, which is desirable in practice." Is this claim because the aprxAns module uses history?
> -    In the perspective of ablation study, this sentence that describes the contribution of AQM+ has two meanings. First, it means that Qgen generates candidate questions considering the context of the dialog history at every turn. Its effect is related to the result of the ablation study on gen1Q. Second, it also means that Qinfo uses aprxAgen, which considers the context of the dialog history (Eq. 3). Its effect is related to the result of the history ablation study.
>
>
> [Minor3]    The RL-QA qualitative results, are these from non-delta or delta? Is there a difference between the two in terms of interpretability?
> -    The RL-QA qualitative results come from delta setting. There is no difference between the two settings in terms of interpretability. Delta setting is just a configuration of hyperparameters where the difference with non-delta setting is that it uses a different weight on one of the loss functions (the model of Das et al. (2017b) optimizes the weighted sum of different loss functions) and a different value for learning rate decay.

---

### Official Review · AnonReviewer3 · 2018-11-02
**This paper addresses the important limitation of the prior work and improves the generalization of the model.**

**Rating:** 6
**Confidence:** 2

**Review:**

The goal of this paper is to build a task-oriented dialogue generation system that can continuously generate questions and make a guess about the selected object.

This paper builds on the top of the previously proposed AQM algorithm and focuses on addressing the limitation of the AQM algorithm, which chooses the question that maximizes mutual information of the class and the current answer, but uses fixed sets of candidate questions/answers/classes.
The proposed AQM+, the extension of AQM, is to deal with 1) the natural language questions / answers using RNN as the generator instead of selecting from the candidate pool (RNN as generator) and 2) a large set of candidate classes (from 10 to 9628).
The novelty is relatively limited, considering that the model is revised from AQM.
Although this work is incremental, this paper addresses the important issue about the generalization.

The experiments show that the model achieves good performance in the experiments.
However, some questions should be clarified.

1) In the ablation study, what is the performance of removing Qpost and remaining Qinfo (asking questions using AQM+, and guessing with an SL-trained model)?

2) In the experiments, the baselines do not contain AQM.
Although AQM has more constraints, it is necessary to see the performance difference between AQM and AQM+, .
If the difference is not significant, it means that this dataset cannot test the generalization capability of the model, so experiments on other datasets may be considered.
If the difference is significant, then the effectiveness of the model is well justified.
The authors should include the comparison in the experiments; otherwise, it is difficult to justify whether the proposed model is useful.

---

> ### Author Response · Authors · 2018-11-17
> **Response to Reviewer 3**
>
> Q.    In the ablation study, what is the performance of removing Qpost and remaining Qinfo (asking questions using AQM+, and guessing with an SL-trained model)?
> -    Thank you for your suggestion. We added the experimental results in Figure 9. According to the result, it seems that AQM+’s Qinfo does not improve the performance of SL’s guesser (Qscore).
> -    For delta setting, we think that the SL guesser is not able to exploit the information from the answers, because the experimental result on SL shows that there is no significant improvement in PMR throughout the dialog in delta setting.
> -    For the non-delta case, it seems that the caption gives dominant information to SL’s guesser rather than dialog history. Thus, questions which often appear with the caption would provide a more clear signal to SL’s guesser for predicting the target class. Figure 9 (a) shows that SL-Q performs better than RL-Q in the early phase, but SL-Q’s performance decreases faster than that of RL-Q in the later phase. We think it is because SL-Q generates the question to be more likely to have co-appeared with the caption than RL-Q. Likewise, it seems that AQM+’s question does not help SL’s guesser because AQM+ generates questions that are more independent of the caption. We also added the description on this ablation study in the paper.
>
>
> Q.    In the experiments, the baselines do not contain AQM.
> -    Fundamentally, it is intractable for previous AQM to deal with GuessWhich due to its large search space. For enabling comparisons, however, we defined several separated AQM-based baselines by replacing each of the components of AQM+ with that of the previous AQM, and then performed ablation studies using those baselines. With these experiments, we empirically showed how much significant our ideas of AQM+ are from the perspective of performance improvement.

---

> > ### Comment · AnonReviewer3 · 2018-11-28
> > **Good paper but the approach is slightly incremental**
> >
> > Thanks for the responses, and I will keep my current score (6; higher than the threshold).
> > The main reason is that the approach is slightly incremental instead of fully original.

---

> > > ### Author Response · Authors · 2018-12-09
> > > **Thank to Reviewer 3**
> > >
> > > We would like to thank the reviewer for considering our responses. We hope that our ablation studies may be able to alleviate the concerns you raised.
> > >
> > > We would like to appeal that the various techniques we suggested to scale up AQM enable the our learning paradigm to be extended further, which is worth sharing with the broader community, including the researchers in multi-modal grounded learning, task-oriented dialog, multi-agent learning, emergent communication, and others.

---

### Official Review · AnonReviewer1 · 2018-11-02
**Contributions seem incremental and concerns regarding the formulated approach**

**Rating:** 6
**Confidence:** 4

**Review:**

The paper proposes an improvement over the AQM approach for an information-theoretic framework for task-oriented dialog systems. Specifically, the paper tries to circumvent the problem of explicitly calculating the information gain while asking a question in the AQM setting. While the original AQM approach sweeps over all possible guesses and answers while estimating information gain, this is rendered impractical in scenarios where this space cannot be tractably enumerated. As a solution, AQM+ proposes sweeping over only some top-k relevant instantiations of answers and guesses in this space by normalizing the probabilities of the subset of the space in consideration. In addition, unlike AQM, AQM+ can ask questions which are relevant to the dialog context so far. Consequentially, this is generalizable and applicable for dialog systems with non ‘yes/no’ answers. Empirical observations demonstrate improvements over the existing approaches for such task-oriented dialog systems. The paper is not very well-written and at times is hard to understand. The contributions seem incremental as well in addition to the concerns mentioned below.

Comments:
- The paper is overloaded with notations and the writing is not very smooth. The terse nature of the content makes it hard to follow in general. If someone apriori was not familiar with task-oriented dialog or the visual dialog setting in Das et al. (2017b), it would be quite hard to follow.
- While mentioning SL/RL approaches while comparing or introducing the setup, the authors do not make any distinction between discriminative and generative dialog models. Specifically, SL approaches could either be trained discriminatively to rank options among the provided ones given dialog context or in a generative manner via token-level teacher forcing. The authors should clearly make this distinction in the introduction and in other places where it’s needed.
- The authors should stress more upon the approximations involved while calculating mutual information. As far as I understand, even in the AQM approach the numerator and the denominator within the logarithm are estimated from a different set of parameters and as such they need not be consistent with each other under marginalization. The term resembles MI and ensuring consistency in such a framework would require either of the numerator or the denominator to be close to something like a variational approximation of the true distribution. In addition, AQM+ adopts the same framework as AQM but computes MI over some top-k of the random variables being considered. Could the authors comment more on why restricting the space of r.v.’s to some top-k samples is a good idea? Would that not lead to somewhat of a biased estimator?
- Unless I am missing something, training aprxAgen from the training data (indA) seems odd. Assuming, this to be Qbot’s mental model of Abot -- there is no prior reason why this should be initialized or trained in such a manner. Similarly, the training paradigm of the depA setting is confusing. If they are trained in a manner similar to a regular Abot -- either SL or RL -- then they’re not approximate mental models but are rather just another Abot agent in play which is being queried by Qbot.
- Under Comparative Models, in paragraph 2 of section 4.1, the authors state that “there are some reports….looks like human’s dialog”. Can the authors elaborate on what they mean by this statement? It’s not clear what the message to be conveyed here is.
- Comparisons in GuessWhich highly rely on the PyTorch implementation in the mentioned github repository. However, the benchmarking performed in that repository for RL over SL is not accurate because of inherent bugs in the implementation of REINFORCE (see https://github.com/batra-mlp-lab/visdial-rl/issues/13 and https://github.com/batra-mlp-lab/visdial-rl/pull/12 ). I would suggest the authors to take this into account.
- Can the authors also show performances for the GuessWhich models (under the AQM+ framework) on the original retrieval metrics for Visual Dialog mentioned in Das et al. (2017a)? This would be useful to judge the robustness of the proposed approach over the methods being compared with.


Updated Thoughts
- The authors adressed the issues raised/comments made in the review. In light of my comments below to the author responses -- I am inclined towards increasing my rating.
- In addition, I have mentioned some updates in the comments which might make the paper stronger -- centered around clarifications regarding the computation of the top-k info-gain term.

---

> ### Author Response · Authors · 2018-11-17
> **Response to Reviewer 1**
>
> We clarified the comments from the reviewer and proofread our paper. Using ablation studies, we empirically showed how much contribution does AQM+ make in the perspective of algorithm, compared to the previous AQM. We also replied to each of your concerns as follows.
>
>
> Q.    The authors should clearly make this distinction between discriminative and generative dialog models.
> -    We explained a distinction between discriminative (retrieval) and generative models (token-level generation) in the revision.
> -    We revised a few expressions and rearranged the citations in the first paragraph, making the first paragraph explain only generative models. We also added the third paragraph to describe a distinction between discriminative and generative models and to explain that the previous AQM, which can be understood as a discriminative model, is extended to a generative model in our AQM+ work.
>
>
> Q.    As far as I understand, even in the AQM approach the numerator and the denominator within the logarithm are estimated from a different set of parameters
> -    The numerator and denominator in the previous AQM (and our AQM+) are estimated from the same set of parameters, which is of the aprxAgen. It is because the denominator (\tilde’{p}) comes from the posterior (\hat{p}) and the numerator (\tilde{p}) (Eq. 6 in our paper), but the posterior comes from the numerator (Eq. 2 in our paper). Please let us know if this does not seem to address the point you made.
>
>
> Q.    Could the authors comment more on why restricting the space of r.v.’s to some top-k samples is a good idea? Would that not lead to somewhat of a biased estimator?
> -    We agree that top-k samples could lead our approximation to be biased toward plausible (high-probability) candidate classes and answers. However, our main goal is to reduce the entropy over plausible candidate classes and answers, not over the whole candidate classes and answers. For this reason, we think our choice is practical for real task-oriented dialog systems. We added this explanation on the manuscript.
>
>
> Q.    The training paradigm of the indA and the depA setting seems odd and confusing.
> -    We follow the perspective and learning paradigm used in the previous AQM paper.
> -    Following the perspective, we argue that the SL algorithm and the indA setting are similar to each other in that Qbot is trained from the training data. Likewise, the RL algorithm and the depA setting are similar in that Qbot is trained from Abot’s responses and the reward of the game. Lee et al. (2018) also argued that the objective function of AQM is similar to that of RL, which links the learning paradigm of AQM with RL.
>
>
> Q.    The authors state that “there are some reports….looks like human’s dialog”. Can the authors elaborate on what they mean by this statement?
> -    It is known that if the distribution of Abot is not fixed during RL, Qbot and Abot can make their own language which is not compatible to natural language (Kottur et al. (2017)). To prevent this problem, many studies added the objective function of language model during RL (Zhu et al. (2017); Das et al. (2017b)). However, Chattopadhyay et al. (2017) reported that fine-tuning both Abot and Qbot with the objective function of RL and language model degrades the communication performance of Abot with human, compared to the pre-trained SL model. According to Lee et al. (2018), this problem comes from the phenomenon that Qbot follows the distribution of Abot implicitly in RL learning. They argue that if the distribution of Abot is changed and becomes far from that of human, then the distribution of Qbot also becomes different from human’s distribution, making the communication performance with human worse. We revised some descriptions of the part the reviewer mentioned.
>
>
> Q.    The benchmarking performance of RL over SL in PyTorch implementation is not accurate because of inherent bugs in the implementation of REINFORCE.
> -    Thank you for letting us know. We will take this issue into account in our research.
> -    Nevertheless, we think this issue on bug does not significantly affect our arguments in the paper. We also compared our algorithm with the results of the original paper (non-delta setting), in not only main experiments but also ablation studies (no caption experiment, QAC experiment). We also conducted additional ablation studies mainly on the non-delta setting in the revision. The scores of SL and RL algorithms under non-delta setting in the paper come from the paper of Das et al. (2017b).

---

> > ### Comment · AnonReviewer1 · 2018-11-30
> > **Comments and updates regarding the author response**
> >
> > Thanks to the authors for providing detailed comments and explanations wherever applicable with respect to the comments provided in the review and updating the paper to reflect the same. In light of the comments below (with respect to the response of the points raised in the review) I am inclined towards increasing my rating for the paper. I will also mention some updates to the revised paper to make the distinction from previous approaches and proposed hypotheses behind the improvements relative to the same more clear which might make the paper stronger.
> >
> > - Thanks for clearly specifying the differences between discriminative and generative models in context of AQM+ and clarifying the reasoning and justification behind several design choices made in the paper. In general, given the revised version, I think the paper is much more easier to follow.
> >
> > - The section highlighting the distinction between AQM+ and the self-play RL approach to the GuessWhich task should explcitly (corresponding to the response mentioned w.r.t. the comment on the earlier version of the statement - “there are some reports… looks like human’s dialog”) should explicitly highlight why AQM+ might not suffer the same consequences unlike the RL setting even with intermediate rewards in the latter paradigm.
> >
> > - It makes sense that restricting MI computation to the top-k samples might render the computation tractable (still biased) -- but it still is not Mutual Information but rather seems like a top-k variant of the Maximum Mutual Information (MMI) criterion. I think the authors should explicitly examine this connection and mention this in the paper. The reason this distinction seems important is because intractable computations of MI in literature have been tackled via variational bounds and since the paper is not doing so -- examining the same and drawing appropriate connections in the paper seem important.

---

> > > ### Author Response · Authors · 2018-12-09
> > > **Thank to Reviewer 1**
> > >
> > > We would like to thank the reviewer for your consideration and further comments. We are glad to hear that the revised manuscript is much easier to follow.
> > >
> > > - The main argument on previous RL algorithms in our paper is that RL did not effectively train the agents for the cooperative game of two machine agents in the setting of the previous works. However, our argument in the statement “there are some reports… looks like human’s dialog” is in a somewhat different context from the main argument. The latter argument implies that even if RL could train their two machine agents effectively for the cooperative game of two agents, the performance on a machine-human game might decrease. This is caused by that the machine agent’s distribution is likely to become far away from the distribution of human’s. We will further clarify this argument in the statement and clearly separate the aforementioned two arguments in our next manuscript.
> > >
> > > - We will add the description on how AQM’s explicit calculation on information gain and posterior has an advantage over the RL, which depends on the capacity of the RNN in the paper.
> > >
> > > - We will add the comparison and connection between mutual information approximation in AQM and comparative neural methods.

---

> ### Author Response · Authors · 2018-11-17
> **Response to Reviewer 1**
>
> Q.     Can the authors also show performances for the GuessWhich models (under the AQM+ framework) on the original retrieval metrics for Visual Dialog mentioned in Das et al. (2017a)?
> -    We used the same Abot model of Das et al. (2017b), which is the same model of Das et al. (2017a). Thus, the performance on the retrieval metrics that the reviewer asked for would be the same as the one reported in Das et al. (2017a). There is no retrieval metric for Qbot in Visual Dialog. As far as we know, PMR is the only available metric for Qbot.
> Please let us know if this does not seem to address the point you made.

---

### Author Response · Authors · 2018-11-17
**For All Reviewers**

We appreciate all reviewers for constructive feedback and comments for the improvement of the paper.

The main contribution of AQM+ lies in its ability to effectively deal with general and complicated task-oriented dialogs where the space of all possible guesses, questions, and answers cannot be tractably enumerated, and thus the previous AQM model is not applicable. Addressing this issue is critical for practical usages on real-world dialogs.
That said, we agree with the feedback of the reviewers that more comparative studies between AQM and AQM+ would be necessary. Hence, to enable AQM to be computationally tractable for GuessWhich, we defined several separated AQM-based baselines by replacing each of the components of AQM+ with that of the previous AQM and then compared the result with AQM+.

AnonReviewer2 (AnonR2) summarized the major departures from the AQM approach claimed in our paper as:
    [1] The generation of candidate questions through beam search rather than predefined set
    [2.1] The approximate answerer being an RNN generating free-form language instead of a binary classifier.
    [2.2] Dropping the assumption that \tilde p(a_t | c, q_t) = \tilde p (a_t | c, q_t, h_{t-1}).
    [3] Estimate approximate information gain using subsets of the class and answer space corresponding to the beam-search generated question set and their corresponding answers.

-    For [1], we performed experiments under the setting where Q_fix, a predefined candidate question set, comes from the generated questions of the SL model. The baseline method with this Q_fix showed significant accuracy degradation (90.76% at |Q|=20 and 92.50% at |Q|=100, indA, non-delta, |A|=|C|=20, and the 10th round) compared to AQM+ (94.64% at |Q|=20). It seems that, in this setting, similar candidate questions highly related to the caption are generated. It results in making the candidate set of questions semantically overlap, and thus degrades the performance. Note that this setting performed even worse than Guesser baseline (92.85%). The result is illustrated in Figure 10.
-    For [1], similar to the above ablation study, we also performed experiments under the setting where Q_fix comes from the training data. The baseline method with this Q_fix showed accuracy degradation (92.79% at indA, non-delta, and the 10th round) compared to AQM+ (94.64%). The result is illustrated in Figure 11. It is noticeable that this setting retrieves questions which are not relevant to the caption nor the target image as can be seen in Figure 13.
-    For [2.1] and [3], we conducted experiments under the setting where the candidate answers are randomly selected from the training data and then fixed. The performance was decreased (from 94.64% to 92.78% at indA, non-delta, and the 10th round). The result is illustrated in Figure 4 (b).
-    For [2.2], we carried out history ablation experiments where the model does not consider the dialog history. The history ablation slightly decreased the performance (from 94.64% to 94.42% at indA, non-delta, and the 10th round). The result is illustrated in Figure 12.
-    For further analysis of AQM+, we investigated how much each of |Q|, |A|, and |C| affects the performance. Decreasing |C| from 20 to 5 decreased the performance from 94.64% to 94.36% at indA, non-delta, |A|=|Q|=20, and the 10th round. We also conducted similar experiments for |Q| and |A|. |Q| affects PMR more, whereas |A| affects PMR relatively less. The results are illustrated in Figure 5.
-    As requested by AnonR3, we conducted experiments under the setting where AQM+’s Qpost and SL’s Qscore are used as the question-generator and the guesser of the model, respectively. AQM+’s Qpost did not increase the performance of SL’s Qscore. It seems that not dialog history but caption information gives dominant information to SL’s guesser. The result is illustrated in Figure 9.

We conducted proofreading. We added tables for terminology to increase the readability. Regarding the issue raised by AnonR1, we revised some descriptions in the introduction section to distinguish between discriminative and generative dialog systems. We also added explanations for the concerns that the reviewers have. Additional improvement on the writing quality and proofreading for the revisions in the rebuttal period will be made until 23 Nov.

---

### Author Response · Authors · 2019-02-21
**Camera Ready Version**

We uploaded the camera ready version of our paper. Also, our code is now publically available: https://github.com/naver/aqm-plus

---

### Meta-Review · Area_Chair1 · 2018-12-14

**Confidence:** 5
**Recommendation:** Accept (Poster)

**Metareview:**

Important problem (visually grounded dialog); incremental (but not in a negative sense of the word) extension of prior work to an important new setting (GuessWhich); well-executed. Paper was reviewed by three experts. Initially there were some concerns but after the author response and reviewer discussion, all three unanimously recommend acceptance.